# A DNA metabarcoding approach for recovering plankton communities from archived samples fixed in formalin

Takuhei Shiozaki[ID][1,2☯]*, Fumihiro Itoh[2☯¤], Yuu Hirose[3], Jonaotaro Onodera[2], Akira Kuwata[4], Naomi Harada[2]

1 Atmosphere and Ocean Research Institute, The University of Tokyo, Kashiwa, Japan, 2 Research Institute for Global Change, Japan Agency for Marine-Earth Science and Technology, Yokosuka, Japan, 3 Department of Applied Chemistry and Life Science, Toyohashi University of Technology, Toyohashi, Japan, 4 Shiogama field station, Fisheries Resources Institute, Japan Fisheries Research and Education Agency, Shiogama, Japan

☯ These authors contributed equally to this work.
¤ Current address: Phytopetrum.Inc, Uruma, Japan
* shiozaki@g.ecc.u-tokyo.ac.jp

**Data Availability Statement:** Although the archived plankton samples in formalin do not exist since they were used for the experiments, all sequence reads are available from the DNA Data

## Abstract

Plankton samples have been routinely collected and preserved in formalin in many laboratories and museums for more than 100 years. Recently, attention has turned to use DNA information from formalin-fixed samples to examine changes in plankton diversity over time. However, no molecular ecological studies have evaluated the impact of formalin fixation on the genetic composition of the plankton community structure. Here, we developed a method for extracting DNA from archived formalin-preserved plankton samples to determine their community structure by a DNA metabarcoding approach. We found that a lysis solution consisting of borate-NaOH buffer (pH 11) with SDS and proteinase K effectively cleaved the cross-link formed by formalin fixation. DNA was extracted from samples preserved for decades in formalin, and the diatom community of the extracted DNA was in good agreement with the microscopy analysis. Furthermore, we stored a plankton sample for 1.5 years and demonstrated that 18S rRNA gene community structures did not change significantly from non-formalin-fixed, time-zero samples. These results indicate that our method can be used to describe the original community structure of plankton archived in formalin for years. Our approach will be useful for examining the long-term variation of plankton diversity by metabarcoding analysis of 18S rRNA gene community structure.

## Introduction

Formaldehyde as a formalin (defined as ~37% formaldehyde solution) has been used widely as a fixative for more than 100 years in research areas from medical and basic biological sciences to environmental sciences [1,2]. These fixed samples are invaluable resources for molecular studies [3], but DNA extraction from formalin-fixed samples is more complicated than that from non-formalin-fixed samples because formalin fixation creates a methylene cross-link

Bank of Japan Sequence Read Archive (accession numbers DRA010173 and DRA010327). All data used to draw the figures are available in the UTokyo Repository (https://doi.org/10.15083/00080064).

**Funding:** This research was supported financially by the Japan Society for the Promotion of Science (JSPS; https://www.jsps.go.jp/english/) KAKENHI Grant JP15H05712 and JP19H05667 to N.H. and JP16H01599 to A.K., Arctic Challenge for Sustainability (ArCS) Project (https://www.nipr.ac.jp/arcs/e/) of the Ministry of Education, Culture, Sports, Science and Technology (MEXT) to N.H., and ArCS II Project (https://www.nipr.ac.jp/arcs2/e/) of the MEXT to T.S. The funders had no role in study design, data collection and analysis, decision to publish, or preparation of the manuscript.

**Competing interests:** The authors have declared that no competing interests exist.

between DNA and protein [1,2], which enables the long-term preservation of organic materials. Therefore, such cross-links must be eliminated prior to DNA extraction.

Methods for extracting and sequencing DNA from formalin-fixed samples have been studied particularly in medical fields, where it is common practice to fix pathological samples in formalin and embed them in paraffin for long-term preservation. The procedure for extracting DNA from formalin-fixed paraffin-embedded (FFPE) samples has been established [4–6], and many companies now offer commercial DNA extraction kits for use with FFPE samples. However, studies on the extraction of DNA from formalin-fixed samples stored in museums and laboratories have just begun [2,7–9]. Museum and laboratory samples differ from FFPE samples in that most of them are stored in buffered formalin solution. Storage in formalin solution results in ongoing cross-linking over time; by contrast, FFPE samples exclude formalin prior to paraffin embedding, so further cross-linking is curtailed. There are some reports of successful extraction and recovery of DNA from samples preserved in formalin for long periods [7–9], but these samples consisted of a single species or group.

We developed a method for extracting DNA from formalin-fixed samples of plankton communities. In oceanography and limnology, plankton communities collected by plankton nets, sediment traps, and continuous plankton recorders have been routinely preserved by formalin-fixation, as opposed to targeting single species [10–12]. To date, such plankton samples have generally been studied with microscopy, which only provides morphological information. Meanwhile, recent molecular metabarcoding techniques have revealed that most planktons are morphologically indistinguishable yet highly diverse [13,14]. DNA information from formalin-fixed plankton samples can be used to identify plankton communities collected in the past, thereby facilitating studies of long-term variation of plankton diversity. Although such attempts have already been made [15,16], no studies have examined whether communities extracted from formalin-fixed samples accurately reflect the communities that were initially preserved. In addition, researchers use their own methods for DNA extraction from formalin-fixed samples, and thus no standard method has been established, as has been done for FFPE samples [5,6]. We tested several lysis solutions and then used the optimal extraction method to investigate changes in the structure of a formalin-preserved plankton community over time with a DNA metabarcoding approach.

## Materials and methods

### Sampling and preparation of formalin-fixed samples

Descriptions of the samples used in this study are summarized in Table 1. Although the samples A–E used were collected by different methods, those were stored in formalin for analysis of the plankton community. All samples except for the control sample S and the time-zero of sample E were fixed with 10% buffered formalin seawater, which was a mixture of formalin buffered with sodium tetraborate (~ pH 8.0) and filtered seawater (unfiltered seawater for sample E) at a ratio of 1:9 [17].

Samples A and B used to evaluate the extract conditions were collected by plankton nets (mesh size 63 μm) during the KH-96-03 cruise of the R/V *Hakuho-maru* in August 1996 and the KT-02-15 cruise of the R/V *Tansei-maru* in May 2002, respectively (n = 1 for each). Samples obtained during the KH-96-03 cruise were collected by surface tows (~5 m) around the Shatsky Rise in the western North Pacific (33.35°N, 159.11°E; sample A). Samples obtained during the KT-02-15 cruise were collected at 70 m by horizontal towing net (MTD net, Rigosha, Tokyo, Japan) [18] in the northwestern North Pacific (39°N, 145°E; sample B). The plankton samples were collected in polypropylene bottles, immediately fixed in 10% buffered

**Table 1. Description of samples used in this study.**

| Name | Sampling tool | Sample type | Region | Location | Date of collection | Depth | Preservation term in formalin | Lysis solutions used[*] |
|------|--------------|-------------|--------|----------|-------------------|-------|------------------------------|------------------------|
| A | Plankton net | Plankton | Temperate | 33.35˚N, 159.11˚E | August, 1996 | ~5 m | 23 years | Nine different lysis solutions |
| B | Plankton net | Plankton | Temperate | 39˚N, 145˚E | May, 2002 | 70 m | 16 years | Nine different lysis solutions |
| C | Sediment trap | Sinking particle | Subarctic | 41.20˚N, 144.68˚E | 21 April–1 May, 2001 | ~1400 m | 17 years | BNB+SDS+ProK |
| D | Sediment trap | Sinking particle | Subarctic | 41.20˚N, 144.68˚E | 11–21 May, 2001 | ~1400 m | 17 years | BNB+SDS+ProK |
| E | Niskin-X | Water | Subtropical | 26.76˚N, 125.59˚E | October, 2017 | 10 m | 0 (not fixed), 0.5, and 1.5 years | BNB+SDS+ProK |
| S[**] | Multiple corer | Marine sediment | Subarctic | 47.00˚N, 160.05˚E | 28 May, 2014 | 5252 m | Not fixed | BNB+SDS+ProK |

[*]Details of lysis solution are written in Table 2.

[**]Sample S was used as a positive control for fluorescent PCR.

formalin, and refrigerated (4 ˚C) until used for experiments (23 and 16 years of storage in formalin, respectively).

Samples C and D were used to evaluate the impact of formalin fixation on the diatom community. Samples C and D were collected in a bottom-tethered sediment trap deployed in the subarctic western North Pacific (41.20˚N, 144.68˚E) from July 2000 to June 2001.

Sediment traps were installed at water depths of ~1400 m to collect sinking particles for 10 days per sample. Before they were deployed, sampling cups were filled with 10% buffered formalin seawater for archiving purposes. We examined sinking particles collected during the spring bloom (April–May). The samples were divided into 10 equal parts using a divider for use in microscopic and DNA analyses, and two of the 10 subsamples were stored at 4 ˚C until analysis. One of the two samples were subjected to DNA extraction for DNA analysis in 2018 after they had been preserved in formalin for 17 years. For the microscopy analyses, diatom cells in the other sample were concentrated in a chamber and counted with an inverted microscope [19]. The identification of diatom species was based on guidelines specified by Hasle and Syvertsen [20].

Sample E was used to examine changes in the entire plankton community during formalin fixation. Sample E was collected in a 20-L polyethylene bag (n = 1) by a Niskin-X bottle during the MR17-07C cruise of the R/V *Mirai* in the subtropical western North Pacific (26.76˚N, 125.59˚E) in October 2017. The seawater sample collected from a water depth of 10 m during the MR17-07C cruise was brought back to the onshore laboratory without any treatment and poured into nine 2-L polypropylene bottles. The first three bottles were taken as samples for time-zero, the next three for 0.5-year incubation, and the remaining three for 1.5-year incubation. The samples for time zero were immediately filtered through 0.22-μm pore size Sterivex filter units (Millipore, Billerica, MA, USA), and the filters were frozen (−80˚C) until analysis. The remaining six samples were mixed with buffered formalin at a ratio of 9:1 and then refrigerated (4˚C). Incubation was terminated by filtration with Sterivex filters after 0.5 or 1.5 years of storage (n = 3 for each).

Sample S, used as a positive control of 18S rRNA gene amplification for fluorescent PCR, was collected on 28 May 2014 in the subarctic western North Pacific (47.00˚N, 160.05˚E). The uppermost layer of the sediment core (a section of 0–2 cm) obtained by a multiple corer was stored at −20˚C without formalin fixation.

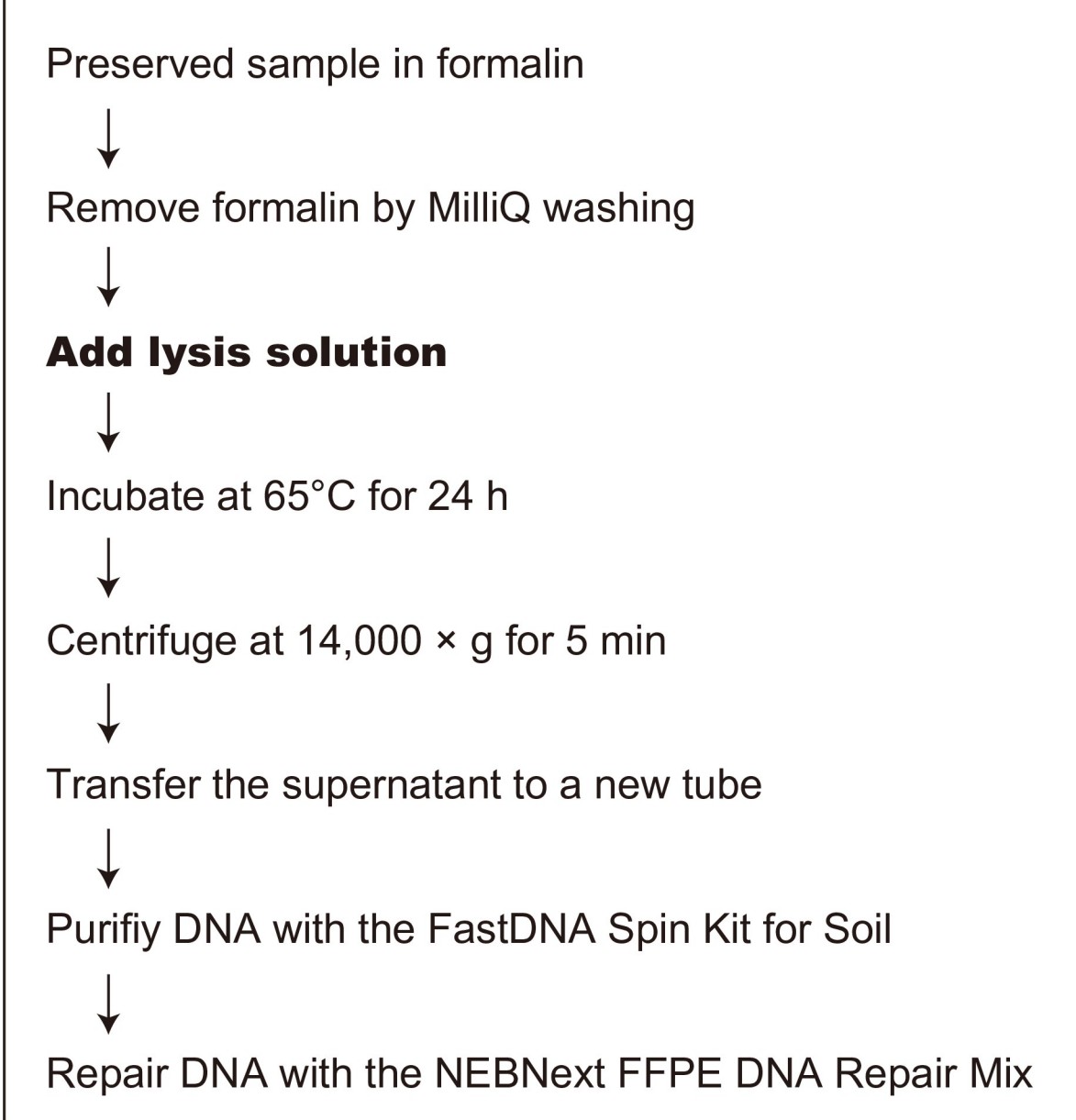

**Fig 1. Procedure for extracting DNA from a sample preserved in formalin.** Nine lysis solutions (see Table 2) were tested in this study (step shown in bold type).

### DNA extraction

Fig 1 illustrates the procedure we developed for extracting DNA from formalin-fixed samples.
Prior to DNA extraction, samples A and B were homogenized by a vortex mixer with 3-mm tungsten carbide beads (Qiagen, Hilden, Germany). Then 0.5 mL of each pellet was transferred into a 1.5-mL tube (n = 3 for each). For samples C–E, the entirety of each sample was used for the experiment. Samples were rinsed with Milli-Q water prior to extraction to remove the formalin.

**Table 2. Lysis solutions tested.**

| No. | Lysis solution |
|-----|----------------|
| 1 | Milli-Q water |
| 2 | 50 mM BNB (pH 11.0) |
| 3 | 50 mM PB (pH 8.5) |
| 4 | 50 mM BNB (pH 11) with SDS* and 50 U ProK |
| 5 | 50 mM PB with SDS* and 50U ProK |
| 6 | 50 mM BNB with SDS*, 50U ProK and dithiothreitol* |
| 7 | 50 mM PB with SDS*, 50U ProK, and dithiothreitol* |
| 8 | 50 mM BNB with SDS*, 50U ProK, and mercaptoethanol* |
| 9 | 50 mM PB with SDS*, 50U ProK, and mercaptoethanol* |

BNB: Borate-NaOH buffer, PB: Phosphate buffer, ProK: Proteinase K.

*The concentration was adjusted to 1% of the premix volume.

DNA was extracted from formalin-fixed samples in two main steps: (1) cleavage of the formalin cross-links and cell lysis, and (2) extraction of the solubilized DNA, with consequent elimination of other cellular components. The first step is important for efficient DNA extraction, and we tested nine lysis solutions in the plankton net samples A and B, as shown in Table 2.

All reagents (except ProK) were premixed, and 980 μL was added to each sample. Both borate-NaOH and phosphate buffer concentrations were 50 mM; SDS and the reducing agent (dithiothreitol or mercaptoethanol) were adjusted to 1% of the premix volume. Recombinant ProK (50U; PCR Grade Solution from *Pichia pastoris*; Roche, Penzberg, Germany) was added to each sample. The manufacturer's recommended incubation time during tissue digestion is 12–18 h; however, longer incubations ($\geq$24 h) are more effective for extracting DNA from formalin-fixed samples [5,21]. Therefore, we set the incubation time to 24 h. Note that a higher incubation temperature facilitates the PCR of formalin-fixed samples [5,22,23]. Thus, we set the incubation temperature to 65˚C, at which ProK is not inactivated [24], although 50˚C is recommended in the manufacturer's instructions.

The subsequent DNA purification step was optimized with the commercially available kit, FastDNA SPIN Kit for Soil (MP Biomedicals, Santa Ana, CA, USA). The samples incubated with lysis solutions were centrifuged at 14,000 × *g* for 5 min. The supernatant was transferred to a new centrifuge tube and mixed with 250 μL of the protein precipitation solution in the kit. The subsequent procedure followed the manufacturer's protocol.

For formalin-fixed samples, DNA repair is recognized to be important after the DNA extraction [5,25]. Thus, we used the NEBNext FFPE DNA Repair Mix (New England Biolabs, Ipswich, MA, USA), developed especially for formalin-fixed samples, to repair DNA damage. The extracted DNA was quantified with a QuantiFluor ONE dsDNA System (Promega, Madison, WI, USA).

The DNA of sample C–E was extracted using a lysis solution containing borate-NaOH buffer (BNB; pH 11.0) with SDS and proteinase K, which was regarded as the optimal lysis solution (see Results and discussion). DNA extraction of sample S was performed using the PowerMax Soil DNA isolation Kit (Mo Bio, Carlsbad, CA, USA).

## Fluorescent PCR

We examined PCR amplifiable DNA with real-time PCR (c.f. [5]). The target region was the V7–V8 region of the eukaryotic 18S rRNA gene, and the primer set used was F-1183

(5′-AATTTGACTCAACACGGG-3′) and R-1631 (5′-TACAAAGGGCAGGGACG-3′) [26]. Fluorescent PCR was performed with StepOne Plus (Applied Biosystems, Foster City, CA, USA). The reaction mixtures (20 μL) contained 10 μL TB Green Premix Ex Taq (Takara Bio, Kusatsu, Japan), 0.2 μM of each primer, 0.4 μL Rox dye, and 2 μL DNA. Sample S, which was not formalin-fixed, was used as a positive control, and distilled water was used as a negative control. All samples were diluted to a uniform concentration of DNA (2 ng μL$^{-1}$). The concentration of DNA extracted with some lysis solutions (MilliQ water [No. 1] of sample A and B and BNB [No. 2] of sample B) was less than 2 ng μL$^{-1}$, in which case the dilution procedure was not performed. Fluorescent PCR was carried out were run in triplicate under the following cycling conditions: 95˚C for 30 s, followed by 45 cycles of 95˚C for 15 s, 56˚C for 30 s, and 72˚C for 30 s, with detection carried out at the end of each cycle. We standardized the Ct value for the positive control and scaled the extent of amplification to each lysis solution by Ct value. The melting curve analysis during the initial optimization of fluorescent PCR revealed that only a single product was produced.

## Metabarcoding and phylogenetic analysis

Because the formalin-fixed samples used in this study were implicitly intended for the preservation of eukaryotic organisms, we amplified and sequenced the 18S rRNA gene for all samples (sample A–E) with a metabarcoding approach as described previously [27]. The V7–V8 region of the 18S rRNA gene was amplified with Tks Gflex DNA Polymerase Low DNA (Takara Bio), using the aforementioned F-1183 and R-1631 primers attached to the adapters Forward (5′-ACACTCTTTCCCTACACGACGCTCTTCCGAT-3′) and Reverse (5′-GTGACTGGAGTTCAGAC GTGTGCTCTTCCGATCT-3′), respectively. PCR was carried out in triplicate for each sample under the following cycling conditions: 94˚C for 2 min, followed by 35 (samples A–D) or 28 (sample E) cycles of 94˚C for 1 min, 54˚C for 1 min, and 72˚C for 1 min, followed by a final extension at 72˚C for 7 min. Triplicate PCR products were pooled and purified with an AMPure XP purification kit (Beckman Coulter, Brea, CA, USA). Index PCR was then performed with a Nextera XT Index kit (Illumina, San Diego, CA, USA) and a KAPA HiFi Hot-Start Ready Mix (KAPA Biosystems, Boston, MA, USA). The PCR products were again purified with the AMPure XP purification kit and quantified with a QuantiFluor ONE dsDNA System. Finally, all samples were mixed in one tube at equimolar concentrations and sequenced by MiSeq (Illumina), wherein 300-bp of each end of the libraries was sequenced with the MiSeq Reagent kit v3 (600 cycles; Illumina) with a PhiX control v3 spike-in (Illumina). Sequenced reads were demultiplexed with MiSeq Reporter v2.6.2 (Illumina). Primer sequences were removed from the sequence reads with Cutadapt [28]. Then the sequence reads were imported into a QIIME 2 program (ver. 2020.2; [29]). Forward and reverse reads were joined, denoised, and checked for chimeras using the DADA2 plug-in [30] with quality filtering thresholds of [6, 8] for the [–p-max-ee-f,–pmax-ee-r] parameters and overlap length thresholds of [250, 200] for the [–p-trunc-len-f,–p-trunc-len-r] parameters. Amplicon sequence variants (ASVs) below 400 bp were filtered out. The ASVs obtained were classified taxonomically with a naïve Bayes classifier trained on reference sequences from SILVA 123 99% operational taxonomic units [31]. The sequence reads used in this study were deposited in the DNA Data Bank of Japan Sequence Read Archive under accession numbers DRA010173 and DRA010327.

Subsequent data analyses were performed with the R platform (ver. 3.5.0; https://www.r-project.org/). Sequence reads were rarefied to the minimum number of reads for sample E. The inverse Simpson (Inv-Simpson) index was calculated to estimate species diversity in each

sample. Differences in community composition for different experimental conditions in sample E were examined by permutational multivariate analysis of variance (PERMANOVA).

## Results and discussion

### Optimal lysis solution for formalin-fixed samples

Nine lysis solutions were tested to lyse the cells and cleave the cross-links formed between the molecules by formalin fixation (Table 2). The amount of DNA extracted from plankton samples A and B (fixed in formalin for 23 and 16 years, respectively) differed greatly among the lysis solutions (Fig 2a). The solutions containing only a buffer of pH 11.0 (borate-NaOH (BNB); No. 2) or 8.5 (sodium phosphate buffer (PB); No. 3) did not substantially enhance DNA yield compared with Milli-Q water (No. 1), although alkalinity has been reported to increase the efficiency of DNA extraction from formalin-fixed samples [22,23,32]. Meanwhile, the solutions with SDS and ProK (Nos. 4–9) markedly increased the DNA yield compared with those that only changed the pH (Nos. 2 and 3; $p < 0.05$). Solutions containing SDS and ProK at pH 11.0 (Nos. 4, 6, and 8) yielded consistently more DNA than solutions at pH 8.5 (Nos. 5, 7, and 9), indicating that the alkaline pH is more effective for DNA extraction.

A reducing agent has been reported to facilitate chemical cleavage of formalin-induced cross-links [5,33–35]; however, there were no significant differences in DNA yield between the samples extracted with solutions containing dithiothreitol (Nos. 6 and 7) or mercaptoethanol (Nos. 8 and 9) and those without a reducing agent (Nos. 4 and 5). We conclude that the use of SDS and ProK under alkaline conditions is most effective for extracting DNA from formalin-fixed samples (Nos. 4, 6, and 8).

To assess the quality of the purified DNA, we performed fluorescent PCR using a universal primer set targeting the eukaryotic 18S rRNA gene. Formalin-fixed samples had significantly increased Ct values compared with the positive control (sample S) (Fig 2b). This indicated that the efficiency of PCR amplification decreased because of remaining damage and/or a methylene cross-link in the template DNA, despite the inclusion of a DNA repair step. Ct values differed slightly among the lysis solutions. The Ct values for BNB with SDS and ProK of sample A (No. 4) and the group containing BNB with SDS and ProK of sample B (Nos. 4, 6, and 8) were lower than in the other lysis solutions. Although the reason for the difference in the trend of the Ct value between samples A and B is unclear, we conclude that DNA extracted using BNB (pH 11) with SDS and ProK was relatively amplifiable by PCR compared with that extracted using other solutions.

In summary, considering the DNA yield obtained and PCR amplification, we conclude that BNB with SDS and ProK (No. 4) was the optimal lysis solution among the ones we studied.

### Structure of the 18S rRNA gene community in samples following long-term preservation in formalin

We performed a metabarcoding analysis of the 18S rRNA gene of samples A–D, which had been archived in buffered formalin for more than 16 years. The number of denoised sequenced reads ranged from 12,169 to 123,259 among the samples, and the rarefaction curve reached a plateau in all samples (S1a Fig).

The plankton community in sample A (plankton net sample fixed for 23 years) was composed mainly of two groups: Metazoa and Rhizaria (Fig 3). This is in good agreement with the composition of surface marine eukaryotic plankton of large size (≥180 μm; [13]). Sample B (plankton net sample fixed for 16 years) was mainly composed of Metazoa and Alveolata (Fig 3). The sampling location, depth, and season differed between samples A and B (Table 1),

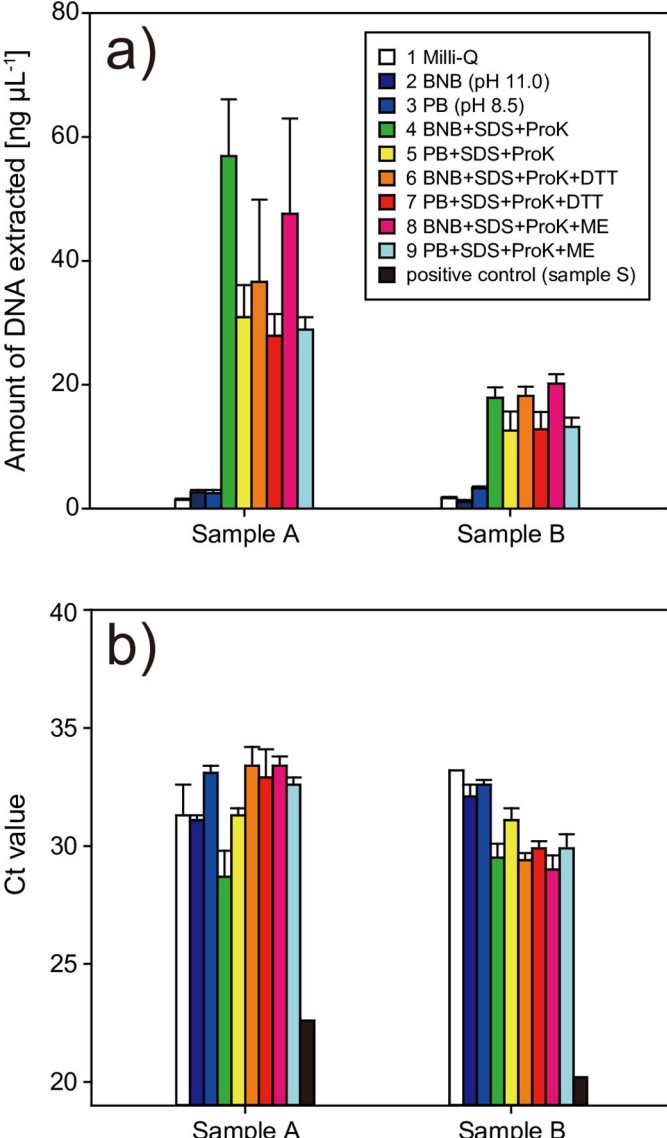

**Fig 2. Amount of DNA extracted and the Ct value for each lysis solution (see Table 2).** a) Amount of DNA extracted. b) The Ct value in the fluorescence PCR.

which may account for the observed difference in plankton composition. The plankton communities of samples C and D collected by sediment traps (fixed for 17 years) were similar, consisting of Metazoa, Fungi, Haptophyta, Rhizaria, Alveolata, and Stramenopiles. However, they differed markedly from those collected with plankton nets (samples A and B; Fig 3). Of these, the Stramenopiles were mostly composed of Diatomea, accounting for 73.9% and 64.4% of the total 18S rRNA gene communities in samples C and D, respectively. In the subarctic western North Pacific where we collected samples C and D, a flux of biogenic silica, which was mainly composed of diatoms, has been reported to account for 60–70% of the total mass flux of sinking particles during the spring bloom period [36], which is consistent with our finding of a high percentage of diatoms in the total community. Taken together, these results suggest that

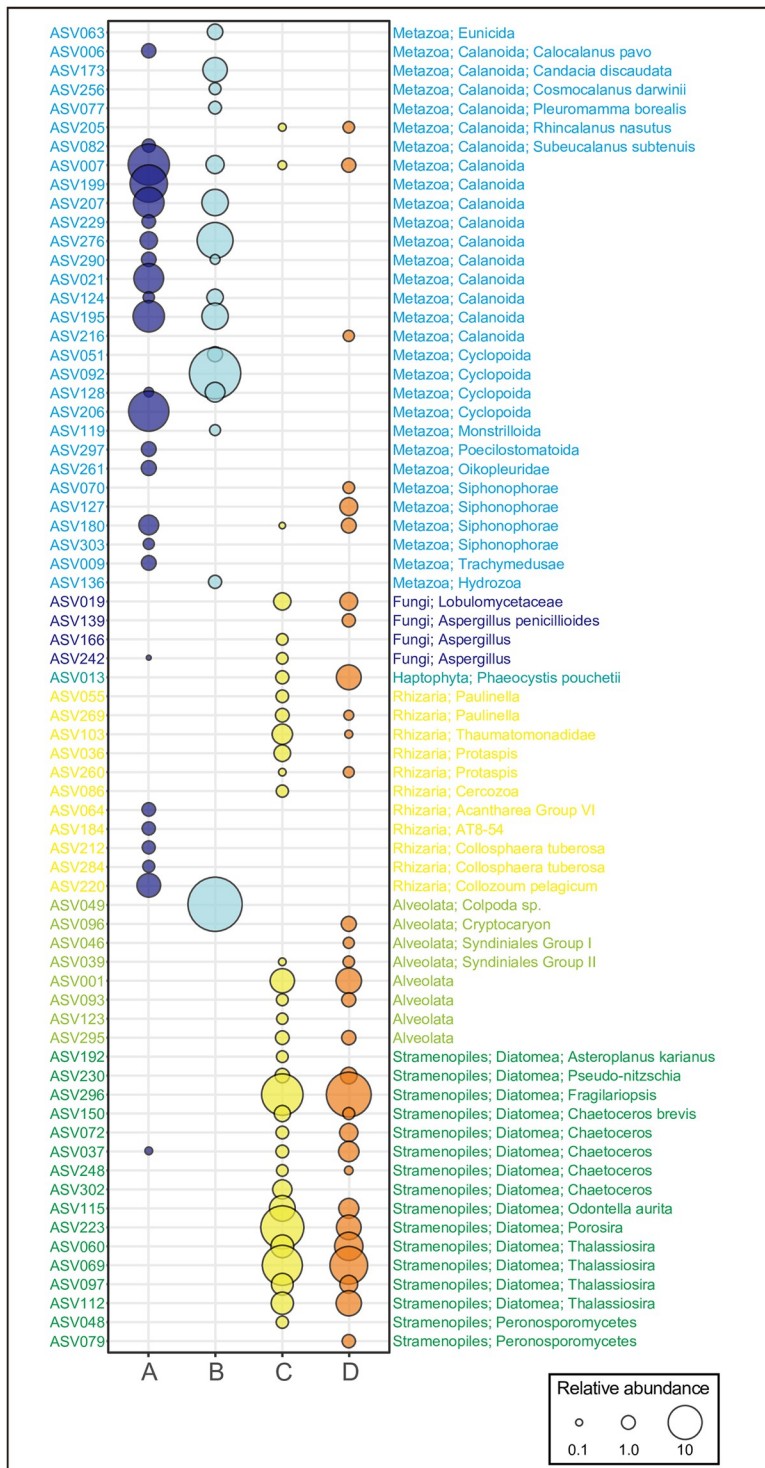

**Fig 3. Composition of the eukaryotic community (≥0.5% of total reads) for samples A, B, C, and D.**

the plankton community structure at a higher taxonomic level can be described based on samples fixed in formalin for a long period.

We further compared the results of microscopy and DNA analyses of diatoms in samples C and D (sediment trap samples fixed for 17 years) (Fig 4). At the genus level, *Thalassiosira*,

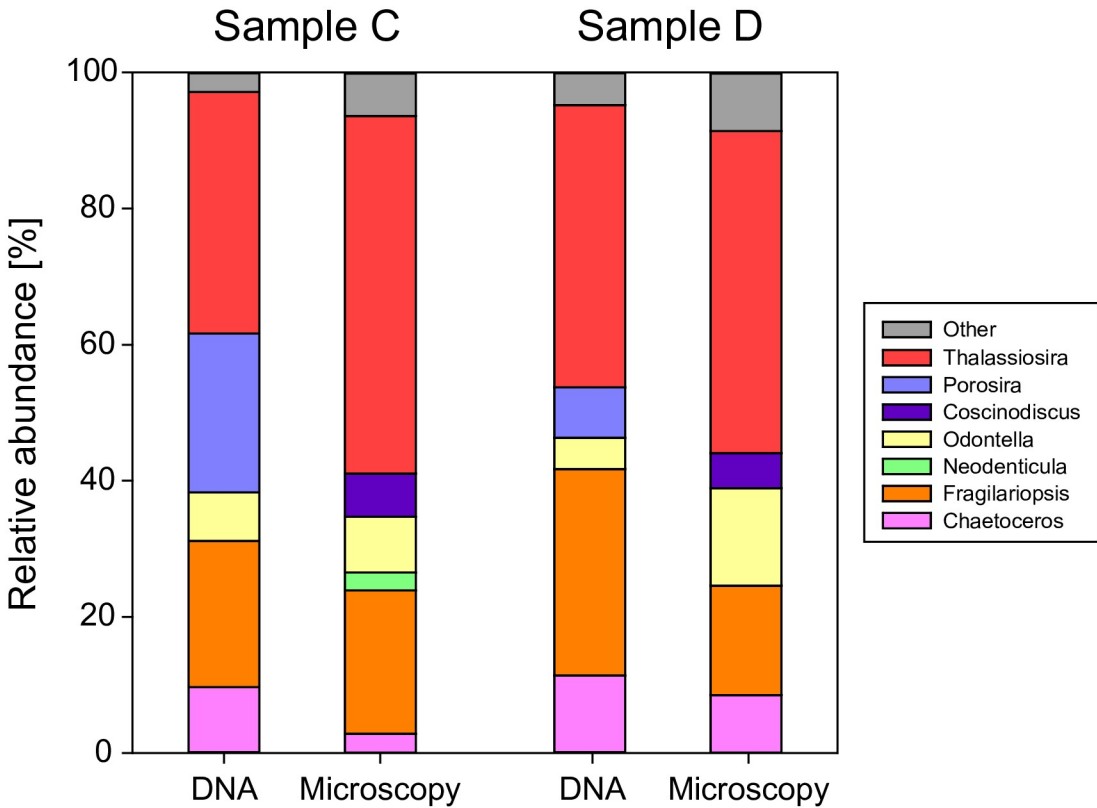

**Fig 4. Comparison of DNA and microscopy analyses of diatoms at the genus level.** In the microscopy analysis, the relative abundance was calculated from the number of diatom cells. Only genera with more than 0.5% of total diatom reads or cell counts are shown. *Porosira* may be misidentified as *Coscinodiscus* in the microscopy analysis (see text).

*Odontella*, *Fragilariopsis*, and *Chaetoceros* were detected by both methods in samples C and D. There were no significant differences between the methods in the overall percentage of these genera. However, some genera appeared only in the microscopy or DNA analysis (*Coscinodiscus*, *Porosira*, and *Neodenticula*). *Porosira* is morphologically similar to *Coscinodiscus* and thus may be misidentified as *Coscinodiscus* based on microscopy [20], or *Porosira* could have been registered incorrectly in the database. Regarding *Neodenticula*, the 18S rRNA gene in the V7–V8 region we targeted is not registered in the database, and thus *Neodenticula* could not be identified from the genome information. Accordingly, we conclude that the diatom compositions of the two methods were very similar. Thus, plankton communities can be identified following our DNA rescue methodology even after long-term preservation in formalin.

### Influence of time on the entire plankton community after formalin fixation

We next examined changes in the plankton community in water sample E during storage in formalin. Sample E was frozen at time zero (without formalin) and stored 0.5 or 1.5 years with formalin. The number of sequence reads after the denoising process ranged from 17,727 to 141,784, and the rarefaction curve reached a plateau in every case (S1b Fig). The major eukaryotic community without formalin fixation was composed of various groups—Metazoa, Apusomonadidae, Picozoa, Telonema, Rhizaria, Alveolata, and Stramenopiles (Fig 5)—which were similar in that they had a small (< 20 μm) marine eukaryotic composition in general [13]. The ASV number and Inv-Simpson index of the time-zero control did not differ significantly from

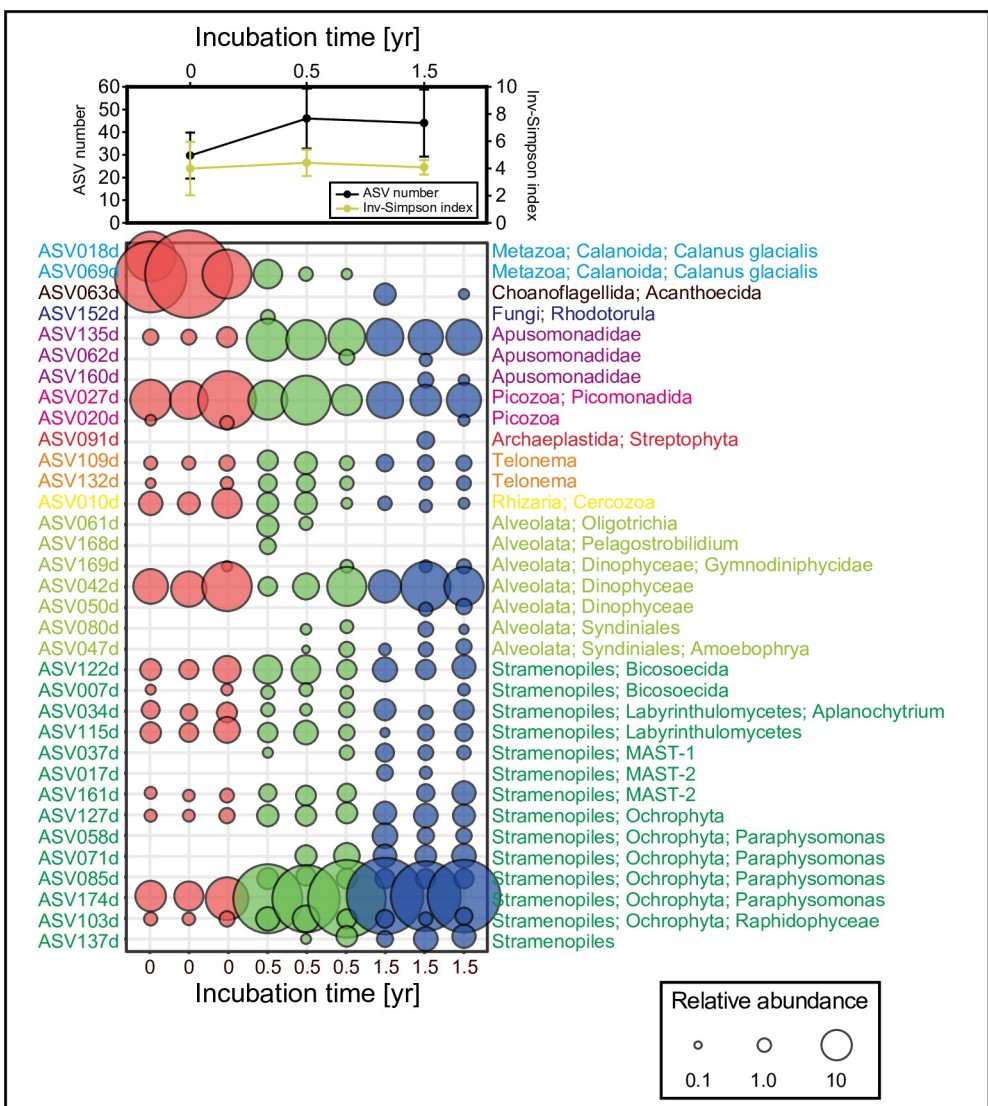

**Fig 5. ASV number, Inv-Simpson index, and composition of the eukaryotic community (≥0.5% of total reads) for sample E for each incubation time.** The area of circles is proportional to the relative abundance.

those of samples stored for 0.5 or 1.5 years ($t$-test, $p > 0.05$; Fig 5), which suggests that the richness and diversity of eukaryotic species can be accurately described after formalin fixation. Furthermore, the composition of the eukaryotic community did not change significantly with 0.5-year or 1.5-year storage compared with the time-zero sample (PERMANOVA, $p > 0.05$). Closer inspection of the ASV table revealed differences in the ASV abundance ratios in formalin-fixed samples relative to time-zero samples. For example, ASV018d and 069d read counts were proportionally higher at time-zero but were rarely observed in formalin-fixed samples. These ASVs belong to the order Calanoida that was also detected in sample A, which had been stored for 23 years in formalin (Fig 3); thus, it is unlikely that DNA could not be recovered from sample E because of formalin fixation. Rather, it could be due to a sampling bias. Calanoid is a motile organism and may have been distributed heterogeneously during the original formalin fixation of the sample. Calanoid may have been dominant at time-zero owing to its large size but was not included in 0.5-year and 1.5-year samples, which may have increased the

relative proportion of other species. A possible explanation for the presence of ASVs detected in formalin-fixed samples but not at time-zero is the replacement of nucleotides as a consequence of formalin fixation [37]. Although such differences may occur through formalin fixation, the results show that the method we developed can be used to retrieve DNA—diverse plankton groups—from formalin-fixed samples.

## Summary and future directions

The present study examined the method of extracting DNA from plankton communities fixed in formalin. According to the DNA yield and the yield for PCR amplification, BNB (pH 11) with SDS and ProK was identified as being the optimal lysis solution from among the ones we studied. We did not examine the DNA purification step, but modifications to this step may further increase DNA yield.

Our results demonstrate that DNA can be extracted not only from single species but also from diverse plankton groups after long-term storage in formalin. All microorganisms are inactivated and fixed immediately after the formalin solution is added (e.g. [38]). Because the chemical structure of DNA is essentially the same in all organisms, information on community structure is also likely to be retained. In fact, the structure of the 18S rRNA gene community did not differ significantly between the 1.5-year formalin-fixed samples and time-zero samples. Moreover, DNA analyses of samples fixed in formalin for 17 years revealed that the diatom community was almost identical to that observed by microscopy. That is, it may be possible to describe the structure of a plankton community based on DNA information, even if samples have been fixed in formalin for a lengthy period.

It should be noted that the samples used in this study covered plankton communities in limited regions and water depths. The diatom community varies greatly depending on the oceanic region [39], but we used samples collected in the subarctic region because the diatoms are relatively abundant and easily identified by microscope. The entire plankton community also varies by region and water depth [13,14], but we used the only samples collected from surface water in the subtropical region for the incubation experiment. Further systematic comparisons are thus required to examine whether DNA can be extracted from more diverse plankton communities in formalin-fixed samples. Also, due to time constraints, we terminated the incubation experiment in 1.5 years. The effect of longer-period formalin storage on individual plankton needs to be examined in more detail.

As mentioned above, some issues still remain for the DNA extraction method from formalin-fixed plankton samples. However, this approach could be a breakthrough in investigating the impact of climate change on plankton biodiversity. Microorganisms, including plankton communities, are greatly affected by environmental changes [40]. However, continuous monitoring for microbial communities has only begun at some fixed observation stations [41,42]. Thus, researchers are only now beginning to understand how human activity and the associated climate change affect microbial communities [43]. Meanwhile, plankton samples have been preserved by formalin fixation in various regions around the world for many years [10–12]. DNA molecules extracted from archived samples of these microbes therefore potentially provide the key to understanding whether and how microorganism communities have been transformed by recent environmental changes globally.

## Supporting information

**S1 Fig. Rarefaction curves for sample a) A, B, C, and D and b) E for each condition.** (PDF)

## Acknowledgments

We thank K. Kimoto, Y. Tada, T. Kitahashi, K. Tadokoro and K. Takahashi for providing valuable samples.

## Author Contributions

**Conceptualization:** Takuhei Shiozaki, Fumihiro Itoh, Naomi Harada.

**Data curation:** Takuhei Shiozaki, Yuu Hirose, Akira Kuwata.

**Funding acquisition:** Takuhei Shiozaki, Akira Kuwata, Naomi Harada.

**Investigation:** Takuhei Shiozaki, Fumihiro Itoh, Yuu Hirose, Jonaotaro Onodera, Akira Kuwata.

**Methodology:** Takuhei Shiozaki, Fumihiro Itoh, Yuu Hirose, Akira Kuwata.

**Project administration:** Naomi Harada.

**Resources:** Takuhei Shiozaki, Jonaotaro Onodera.

**Supervision:** Naomi Harada.

**Validation:** Takuhei Shiozaki, Yuu Hirose, Akira Kuwata.

**Visualization:** Takuhei Shiozaki, Yuu Hirose, Akira Kuwata.

**Writing – original draft:** Takuhei Shiozaki.

**Writing – review & editing:** Takuhei Shiozaki, Yuu Hirose, Jonaotaro Onodera, Akira Kuwata, Naomi Harada.

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
