## [Decision Letter · Decision Letter 0]

3 Dec 2020

PONE-D-20-34778

A DNA metabarcoding approach for recovering plankton communities from archived samples fixed in formalin

PLOS ONE

Dear Dr. Shiozaki,

Thank you for submitting your manuscript to PLOS ONE. After careful consideration, we feel that it has merit but does not fully meet PLOS ONE’s publication criteria as it currently stands. Therefore, we invite you to submit a revised version of the manuscript that addresses the points raised during the review process.

The two reviews obtained indicate the contribution useful and have made comments that need your consideration and appropriate incorporation.

We look forward to receiving your revised manuscript.

Kind regards,

Arga Chandrashekar Anil, Ph. D., D. Agr.,

Academic Editor

PLOS ONE

Journal Requirements:

2. Please can you clarify if it is possible for other researchers to access the samples used in your study for the purposes of reproducing your analyses.

If so, please provide details in your materials and methods section of how the samples can be accessed.

'This research was supported financially by the Japan Society for the Promotion of Science (JSPS; https://www.jsps.go.jp/english/) KAKENHI Grant JP15H05712 (N.H.), Arctic Challenge for Sustainability (ArCS; https://www.nipr.ac.jp/arcs/e/) (N.H.), and ArCS II (https://www.nipr.ac.jp/arcs2/e/)(T.S.) Projects. The funders had no role in study design, data collection and analysis, decision to publish, or preparation of the manuscript.'

We note that one or more of the authors are employed by a commercial company: Phytopetrum.Inc, Japan

Reviewers' comments:

Reviewer's Responses to Questions

**Comments to the Author**

1. Is the manuscript technically sound, and do the data support the conclusions?

Reviewer #1: Yes

Reviewer #2: Yes

2. Has the statistical analysis been performed appropriately and rigorously? 

Reviewer #1: Yes

Reviewer #2: Yes

3. Have the authors made all data underlying the findings in their manuscript fully available?

Reviewer #1: Yes

Reviewer #2: Yes

4. Is the manuscript presented in an intelligible fashion and written in standard English?

Reviewer #1: Yes

Reviewer #2: Yes

5. Review Comments to the Author

Reviewer #1: Manuscript ID: PONE-D-20-34778

Reviewer’s comments

In this manuscript by Shiozaki et al entitled “A DNA metabarcoding approach for recovering plankton communities from archived samples fixed in formalin”, the authors developed a method for extracting DNA from archived formalin-preserved plankton samples to determine their community structure by a DNA metabarcoding approach and indicated its use to describe the original community structure of plankton stored in formalin for years. This approach is useful for examining the long-term variation of planktons by metabarcoding analysis of 18S rRNA gene community structure. The manuscript is relatively well written but needs improvement as there are many grammatical errors throughout the manuscript. In my opinion, this paper has a good potential to be published in the journal and can be accepted after minor revision. My concerns and suggestions are as follows:

The introduction section is well written. In the methods section, my main concern is the lack of description of some key aspects of the experimental design such as the number of replicates analysed, description of samples etc. that prevent assessment of the statistical soundness of the results. The methods need to be made clearer. In view of this, a schematic on the experimental design emphasizing on the sampling and different extraction protocols will be useful. Also, since the focus of this paper is on metabarcoding analysis of 18S rRNA gene community structure it would be useful to provide details on composition of the eukaryotic community in the main manuscript instead of including in the supplementary file 2. The rational for using sample S, which was not formalin fixed as a positive control and not sample E (time zero) which was also not fixed with formalin need to be provided. In the case of results section, especially for long-term preservation in formalin, it will be helpful if the authors refer to the samples with respect to age (plankton samples that were preserved for 16 and 23 years and the sediment trap samples stored for 17 years) or type (plankton/sediment) instead of codes such as ABCD as it is confusing. Also, when comparing the results of the microscopy and DNA analysis, the interpretation’s provided for the discrepancies has to be realistic.

Besides, the sub-title Time-variation in an entire plankton community after formalin fixation may be replaced with “Influence of time on the entire plankton community after formalin fixation”.

Reviewer #2: The paper is nicely written and easy to follow. The authors test how formalin fixation affects the DNA content of plankton samples that have been stored for many years. They test nine different DNA extraction methods on samples stored in formalin for up to 23 years. They also provide a comparison of changes in plankton community for samples collected at the same time and place where the DNA was extracted immediately for one sample and stored for 0.5 and 1.5 years for two additional samples. They examine the success of DNA extraction using real-time PCR, metabarcoding of the V7-V8 region of the 18S rRNA gene as well as traditional cell counting of diatoms. Their findings can be useful for other researchers who are working with samples preserved in formalin.

Some minor comment:

L21: “tuned to” should be “turned to”.

L34: “have been examined to evaluate”, change to “have evaluated”.

L35: “planktons” should be “plankton diversity”

L242: I can’t see “PC” in figure 2.

L299 (and elsewhere): The authors use SV as an abbreviation for the sequence variants from Dada2. However, the majority of publications use ASV, short for amplicon sequence variation. ASV is also the term preferred by the authors of Dada2.

Figure 2: There is no title in the small boxed legend in the figure (numbered 1-9 + std) . I guess it refers to lysis solution in table 2. The figure would be easier to read if the authors could fit a small description of the lysis buffer next to the number. (E.g. “1: BNB, pH 11” for the first “2: PB, pH 8.5” for the second, and so on).

Figure 3: I very much prefer supplemental figure S2 to the pie charts used in this figure. I suggest that authors make figure S2 slimmer by removing the full taxonomic assignment for the SVs (which should be renamed ASVs), and keep only the final part of the name (i.e. Phaeocystis poechetii, Eunicida sp., Calocalanus pavo, etc). The color scheme can be kept to separate between different supergroups. For the colors I would also differentiate between Metazoa and Fungi. Finally, I think it makes more sense to put the haptophytes underneath the fungi, together with SAR. Then S2 could probably be the main Figure 3..

Figure 5. The names can be shortened as described for the previous figure. Also I the ASVs it could be ordered more logically from top to bottom: Metazoa, Choanoflagellates, Fungi, Apusomonadidae, Picozoa, Archaeplastida, Telonema, Rhizaria, Alveolata, Stramenopiles (see Burki et al. 2019, The New Tree of Eukaryotes, and Adl et al 2019 Revisions to the Classification, Nomenclature, and Diversity of Eukaryotes)

6. PLOS authors have the option to publish the peer review history of their article (what does this mean?). If published, this will include your full peer review and any attached files.

Reviewer #1: No

Reviewer #2: No

---

## [Author Response · Author response to Decision Letter 0]

22 Dec 2020

Reviewer #1: Manuscript ID: PONE-D-20-34778

Reviewer’s comments

In this manuscript by Shiozaki et al entitled “A DNA metabarcoding approach for recovering plankton communities from archived samples fixed in formalin”, the authors developed a method for extracting DNA from archived formalin-preserved plankton samples to determine their community structure by a DNA metabarcoding approach and indicated its use to describe the original community structure of plankton stored in formalin for years. This approach is useful for examining the long-term variation of planktons by metabarcoding analysis of 18S rRNA gene community structure. The manuscript is relatively well written but needs improvement as there are many grammatical errors throughout the manuscript. In my opinion, this paper has a good potential to be published in the journal and can be accepted after minor revision. My concerns and suggestions are as follows:

The introduction section is well written. In the methods section, my main concern is the lack of description of some key aspects of the experimental design such as the number of replicates analysed, description of samples etc. that prevent assessment of the statistical soundness of the results. The methods need to be made clearer. In view of this, a schematic on the experimental design emphasizing on the sampling and different extraction protocols will be useful. 

 We have corrected grammatical errors throughout the manuscript.

 We have added the number of samples used in the experiments at L88, 104-107, and 110.

 The only difference in the extraction protocol is the lysis solution used. Thus, we have added which lysis solution was used for each sample in Table 1.

Also, since the focus of this paper is on metabarcoding analysis of 18S rRNA gene community structure it would be useful to provide details on composition of the eukaryotic community in the main manuscript instead of including in the supplementary file 2. 

 We have moved Figure S2 to the main text.

The rational for using sample S, which was not formalin fixed as a positive control and not sample E (time zero) which was also not fixed with formalin need to be provided. 

 This is a practical reason. DNA extraction of sample E had not yet been performed when the fluorescent PCR experiment was conducted.

In the case of results section, especially for long-term preservation in formalin, it will be helpful if the authors refer to the samples with respect to age (plankton samples that were preserved for 16 and 23 years and the sediment trap samples stored for 17 years) or type (plankton/sediment) instead of codes such as ABCD as it is confusing. 

 We have added preservation term and sample type in the result section (L231, 272, 275, 278, 295, and 316).

Also, when comparing the results of the microscopy and DNA analysis, the interpretation’s provided for the discrepancies has to be realistic.

 The differences between the two methods are Coscinodiscus, Porosira, and Neodenticula. As mentioned in the manuscript, this difference can be 

 explained by problems with the DNA database and misidentification by microscopy.

Besides, the sub-title Time-variation in an entire plankton community after formalin fixation may be replaced with “Influence of time on the entire plankton community after formalin fixation”.

 We have changed as suggested.

 

Reviewer #2: The paper is nicely written and easy to follow. The authors test how formalin fixation affects the DNA content of plankton samples that have been stored for many years. They test nine different DNA extraction methods on samples stored in formalin for up to 23 years. They also provide a comparison of changes in plankton community for samples collected at the same time and place where the DNA was extracted immediately for one sample and stored for 0.5 and 1.5 years for two additional samples. They examine the success of DNA extraction using real-time PCR, metabarcoding of the V7-V8 region of the 18S rRNA gene as well as traditional cell counting of diatoms. Their findings can be useful for other researchers who are working with samples preserved in formalin.

 We appreciate for your constructive comments. We have corrected the points you raised as follows.

Some minor comment:

L21: “tuned to” should be “turned to”.

 Changes as suggested.

L34: “have been examined to evaluate”, change to “have evaluated”.

 Changed as suggested.

L35: “planktons” should be “plankton diversity”

 Changed as suggested.

L242: I can’t see “PC” in figure 2.

 We have reworded “positive control” in the revised figure.

L299 (and elsewhere): The authors use SV as an abbreviation for the sequence variants from Dada2. However, the majority of publications use ASV, short for amplicon sequence variation. ASV is also the term preferred by the authors of Dada2.

 We have changed the wording from SV to ASV throughout the manuscript.

Figure 2: There is no title in the small boxed legend in the figure (numbered 1-9 + std) . I guess it refers to lysis solution in table 2. The figure would be easier to read if the authors could fit a small description of the lysis buffer next to the number. (E.g. “1: BNB, pH 11” for the first “2: PB, pH 8.5” for the second, and so on).

 We have added the information of the lysis buffer in the legend.

Figure 3: I very much prefer supplemental figure S2 to the pie charts used in this figure. I suggest that authors make figure S2 slimmer by removing the full taxonomic assignment for the SVs (which should be renamed ASVs), and keep only the final part of the name (i.e. Phaeocystis poechetii, Eunicida sp., Calocalanus pavo, etc). The color scheme can be kept to separate between different supergroups. For the colors I would also differentiate between Metazoa and Fungi. Finally, I think it makes more sense to put the haptophytes underneath the fungi, together with SAR. Then S2 could probably be the main Figure 3.

 We have moved Figure S2 to the main text, made figure slimmer, and changed the order as suggested.

Figure 5. The names can be shortened as described for the previous figure. Also I the ASVs it could be ordered more logically from top to bottom: Metazoa, Choanoflagellates, Fungi, Apusomonadidae, Picozoa, Archaeplastida, Telonema, Rhizaria, Alveolata, Stramenopiles (see Burki et al. 2019, The New Tree of Eukaryotes, and Adl et al 2019 Revisions to the Classification, Nomenclature, and Diversity of Eukaryotes)

 We have changed the order as suggested.

---

## [Editor Report · Decision Letter 1]

11 Jan 2021

A DNA metabarcoding approach for recovering plankton communities from archived samples fixed in formalin

PONE-D-20-34778R1

Dear Dr. Shiozaki,

We’re pleased to inform you that your manuscript has been judged scientifically suitable for publication and will be formally accepted for publication once it meets all outstanding technical requirements.

Kind regards,

Arga Chandrashekar Anil, Ph. D., D. Agr.,

Academic Editor

PLOS ONE
---

## [Editor Report · Acceptance letter]

26 Jan 2021

PONE-D-20-34778R1 

A DNA metabarcoding approach for recovering plankton communities from archived samples fixed in formalin 

Dear Dr. Shiozaki:

I'm pleased to inform you that your manuscript has been deemed suitable for publication in PLOS ONE. Congratulations! Your manuscript is now with our production department. 

Kind regards, 

on behalf of

Professor Arga Chandrashekar Anil 

Academic Editor

PLOS ONE